# Rational design of ¹⁹F NMR labelling sites to probe protein structure and interactions

**Julian O. Streit** [1,2] ✉, **Sammy H. S. Chan** [1,2] ✉, **Saifu Daya**[1] & **John Christodoulou** [1] ✉

Proteins are investigated in increasingly more complex biological systems, where ¹⁹F NMR is proving highly advantageous due to its high gyromagnetic ratio and background-free spectra. Its application has, however, been hindered by limited chemical shift dispersions and an incomprehensive relationship between chemical shifts and protein structure. Here, we exploit the sensitivity of ¹⁹F chemical shifts to ring currents by designing labels with direct contact to a native or engineered aromatic ring. Fifty protein variants predicted by AlphaFold and molecular dynamics simulations show 80–90% success rates and direct correlations of their experimental chemical shifts with the magnitude of the engineered ring current. Our method consequently improves the chemical shift dispersion and through simple 1D experiments enables structural analyses of alternative conformational states, including ribosome-bound folding intermediates, and in-cell measurements of protein-protein interactions and thermodynamics. Our strategy thus provides a simple and sensitive tool to extract residue contact restraints from chemical shifts for previously intractable systems.

NMR spectroscopy is a powerful experimental technique to probe biomolecular structure, interactions, and dynamics across different timescales. As a deeper understanding of protein structure within larger complexes and its physiological environment is increasingly being sought, ¹⁹F NMR has re-emerged as a valuable tool in contemporary structural biology, with continuing methodological advancements permitting the study of systems that are inaccessible by other means[1–5].

The fluorine nucleus is highly attractive for biomolecular NMR, as its 100% natural abundance and high gyromagnetic ratio (94% of ¹H) ensures high (signal-to-noise) sensitivity, while it is inexpensive relative to other labelling schemes. Residue-type or site-specific labelling by biosynthetic means or covalent modification results in background-free labelling and reduced spectral complexity that permits the use of simple 1D pulse-acquire experiments. Thus, lower sample concentrations are required, with greater spectral observability of larger molecules and challenging systems, including membrane proteins[2,3,5–7], nascent polypeptides in complex with their ~2.5-MDa parent

ribosome[8–10], and within living bacterial[11,12] and mammalian cells[13,14]. Fluorine NMR is also commonly used in drug screening[15].

The use of trifluoromethyl (CF₃) groups has become the preferred reporter for proteins because of the high sensitivity from the three-fold degeneracy and reduced chemical shift anisotropy (CSA) from the rotationally mobile CF₃ group[16]. However, this also results in limited protein chemical shift dispersions of only ~2 ppm for membrane proteins[2–5], deriving from its reduced sensitivity to local electric fields and Van der Waals interactions compared to monofluorinated tags[3,17], though new fluorinated amino acids are being developed to improve chemical shift sensitivity[3,17,18]. The limited CF₃ group chemical shift dispersion and coinciding line broadening (associated with fast spin relaxation of slow tumbling, large biomolecules) can reduce the ability to resolve multiple conformational states or labelling sites[2–4,8,17,19]. These effects are exacerbated in cells by quinary interactions[20–22] and decelerated diffusion (where linewidths of up to ~1 ppm have been reported[11]).

A second restriction is that, currently, ¹⁹F NMR is not routinely used to obtain structural restraints. Unlike other NMR nuclei whose chemical

[1]Institute of Structural and Molecular Biology, University College London and Birkbeck College, London, UK. [2]These authors contributed equally: Julian O. Streit, Sammy H. S. Chan. ✉e-mail: julian.streit.16@ucl.ac.uk; s.chan.12@ucl.ac.uk; j.christodoulou@ucl.ac.uk

shifts can be used to derive structural models[23-26], the origins of [19]F protein chemical shifts are not well-understood[27-29], and predictions rely on computationally expensive quantum chemical calculations[30]. Moreover, like many other methods, such as Förster resonance energy transfer (FRET) and double electron-electron resonance (DEER), [19]F NMR requires protein modification, and the choice of labelling site(s) is a key consideration. [19]F labels are, however, typically less bulky than the modifications used for FRET or DEER, with analogous fluorinated amino acids available[29] and, despite not being routinely used as structural restraints, their incorporation provides opportunities to relate site-specific [19]F chemical shifts to local protein structure.

We present a design strategy that uses the commercially available [19]F-label 4-trifluoromethyl-L-phenylalanine (tfmF) to help overcome challenges of chemical shift dispersion and structural interpretation. Motivated by previous reports of aromatic ring currents affecting [19]F NMR chemical shifts[5,6,31], we have rationally designed 50 protein variants with fluorine labelling sites near native or engineered aromatic rings. Using AlphaFold[32-35] and all-atom molecular dynamics (MD) as predictors with 80-90% success, Van der Waals contacts between two sidechains are measured directly as the [19]F chemical shift from simple 1D NMR spectra. We illustrate applications of our approach to multiple protein systems spanning the folding of isolated proteins, large macromolecular complexes, ligand interactions and in-cell protein-protein interactions where other experimental methods are unable to provide residue-specific insights of the dynamic interconversions that are the mainstay of biological function.

## Results

### Ring current effects improve [19]F chemical shift dispersion

To begin exploring the utility of probing different sites by [19]F NMR, we used the model immunoglobulin-like domain FLN5, whose structure and folding in vitro[36,37] and co-translational folding (coTF) on the ribosome[8,9,38,39] has been well-characterised by NMR spectroscopy,

including by [19]F-labelling[8,10]. We initially chose [19]F-labelling using the non-canonical amino acid tfmF because it yields NMR spectra with high signal-to-noise for even MDa complexes[8], is commercially available, and can easily be incorporated into standard bacterial strains[40]. Similar fluorinated phenylalanine analogues have also been incorporated in human cells[13,41]. An efficient in-frame amber suppression protocol (> 95% incorporation[8]) was used to incorporate tfmF at genetically specified positions in FLN5 (Fig. 1a, b) in E. coli. We produced 18 variants of folded FLN5, each tfmF-labelled at different positions across the protein structure (16 solvent-exposed, and 2 within the hydrophobic core, 691tfmF and 747tfmF) and all yielding a single resonance in their [19]F NMR spectrum. The chemical shifts of the 18 labelling sites ranged across ~ 0.4 ppm (Fig. 1c), centred at the random coil value (61.82 ppm as observed for three tfmF-labelling sites on an unfolded variant of FLN5, Fig. 1c), and thus appear to be significantly narrower in their dispersion compared to the ~ 2 ppm chemical shift range of trifluoromethyl-labelled membrane proteins[2-5], likely due to a more uniform chemical environment in bulk solution compared to membranes.

Two exceptions to this narrow dispersion of chemical shifts are 655tfmF (- 62.61 ppm) and 732tfmF (- 61.44 ppm). Inspection of the FLN5 crystal structure revealed that both residues are uniquely positioned near residues with a (de)shielding effect. 732tfmF is positioned close to a negatively charged Glu692 on the neighbouring strand, which likely explains the observed deshielding effect (+ 0.38 ppm) as protein [19]F chemical shifts have been shown to be sensitive to charge mutations, although the magnitude of electrostatic interactions contributing to shielding remains poorly understood[28]. A larger dispersion was observed for 655tfmF, exhibiting a resonance - 0.79 ppm from the random coil. Because residue Tyr655 is in close contact with Phe675 on the adjacent β-strand (4.3 Å between OH atom and ring centre of mass, Fig. 1d), we tested whether a ring current effect induced by Phe675 could cause shielding of the [19]F nucleus of 655tfmF. Mutation of

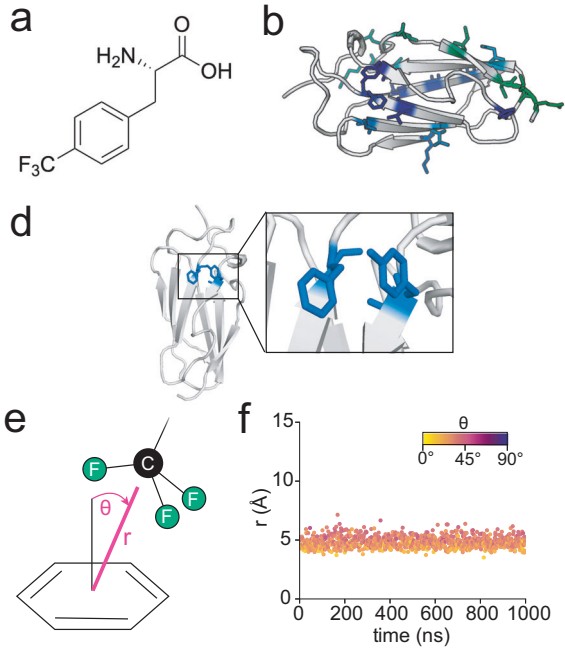

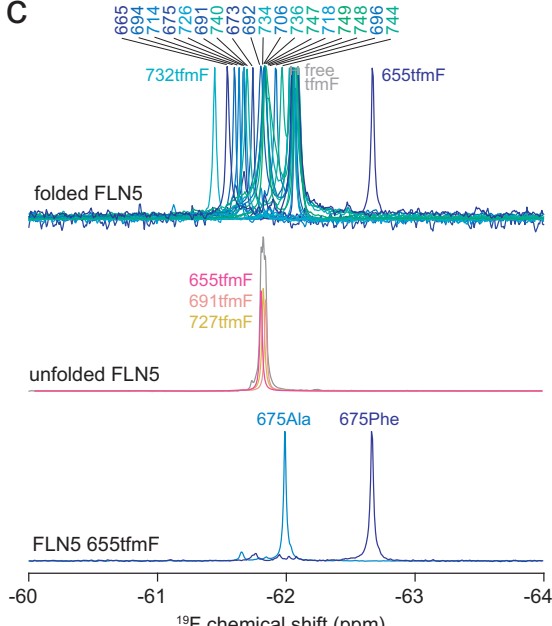

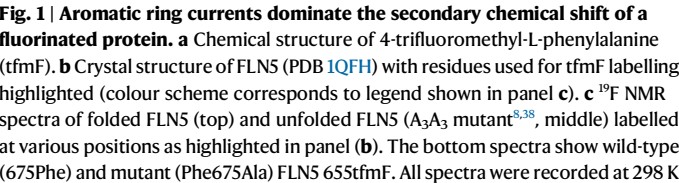

**Fig. 1 | Aromatic ring currents dominate the secondary chemical shift of a fluorinated protein. a** Chemical structure of 4-trifluoromethyl-L-phenylalanine (tfmF). **b** Crystal structure of FLN5 (PDB 1QFH) with residues used for tfmF labelling highlighted (colour scheme corresponds to legend shown in panel **c**). **c** [19]F NMR spectra of folded FLN5 (top) and unfolded FLN5 (A3A3 mutant[8,38], middle) labelled at various positions as highlighted in panel (**b**). The bottom spectra show wild-type (675Phe) and mutant (Phe675Ala) FLN5 655tfmF. All spectra were recorded at 298 K

and 500 MHz. **d** Close-up view of Tyr655 and Phe675 in the crystal structure of FLN5. **e** Schematic illustration of a trifluoromethyl (CF3) group interacting with an aromatic benzene ring, defined by the distance, r, between the CF3 and ring centres of mass and the angle, θ, between the vector normal to the ring plane and the vector between the CF3 and ring centres of mass. **f** All-atom MD simulation of FLN5 655tfmF showing an interaction between the CF3 group of 655tfmF and the aromatic ring of Phe675 quantified using r and θ.

Phe675 to alanine resulted in only small $^1$H/$^{15}$N amide backbone CSPs, with the protein remaining folded (Supplementary Fig. 1a). However, we observed a significantly less shielded $^{19}$F chemical shift of -62.0 ppm, within the range of all other labelling sites described above (Fig. 1c). Ring current effects therefore appear to dominate the secondary chemical shifts ($\Delta\delta = \delta_{observed} - \delta_{random\ coil}$) of solvent-exposed fluorine labelling sites, and the resulting chemical shifts are thus direct reporters of specific sidechain contacts.

To quantify the sidechain interactions and thus ring current effects between 655tfmF and 675Phe, we used all-atom molecular dynamics (MD) simulations with two force fields, ff15ipq[42,43] and C36m[44,45] (see "Methods"). Simulations with both force fields corroborate a stable sidechain interaction, with an average distance of ~ 5 Å and a perpendicular orientation between the CF$_3$ group and the benzyl ring of 675Phe, collectively compatible with a shielded ring current effect as measured experimentally (Fig. 1e, f and Supplementary Fig. 1b, c). In the reverse scenario, with 675tfmF and Tyr655, we observe distances that are not compatible with ring current effects (> 7 Å, Supplementary Fig. 1b, c), consistent with an experimentally observed $^{19}$F chemical shift of -61.67 ppm (deshielded by 0.15 ppm relative to random coil, Supplementary Table 3). The 655tfmF ring current interaction is also independent of the initial sidechain orientation of 655tfmF (Supplementary Fig. 1b, c), suggesting these simulations can be predictive.

We also assessed site-specific labelling on FLN5 at position 655 using two alternative trifluoromethyl probes. Incorporation of 4-(trifluoromethoxy)-L-phenylalanine (OCF$_3$Phe) by amber suppression yields NMR resonances with high signal-to-noise, but with a chemical shift dispersion reduced from that of tfmF by 0.56 ppm (70%, Supplementary Fig. 2). Similarly, post-translational cysteine modification with 2-bromo-N-(4-[trifluoromethyl]phenyl)acetamide (BTFMA) has been shown to be most sensitive to solvent polarity (and likely protein environments) among trifluoromethyl tags[19], yet for FLN5 Tyr655Cys Cys747Val results in only a < 0.01 ppm secondary chemical shift (Supplementary Fig. 3a). The increased flexibility and longer sidechains of both OCF$_3$Phe and Cys-BTFMA are likely incompatible with close Phe675 contacts to induce a strong ring current effect (Supplementary Fig. 3b–d), and we therefore further explored the use of tfmF.

### Rational design of ring current shifts to probe protein structure

We explored whether solvent-exposed fluorine labelling sites could be designed with structurally interpretable and enhanced secondary chemical shifts using native or engineered ring currents. Given the precise nature of a ring current contact and the time-consuming effort of an entirely trial-and-error experimental approach, we instead developed a rational design approach exploiting either native aromatics or engineered residues (Fig. 2a), where we used Phe and His substitutions for engineered residues due to their smaller sizes compared to Tyr and Trp. We initially applied this approach to FLN5 and human HRAS. We predicted and quantified sidechain orientations of new tfmF/aromatic-residue labelling pairs using all-atom MD simulations performed in triplicate from different initial sidechain orientations (3 × 1 μs, see Methods, Supplementary Table 1 and Supplementary Fig. 4), and orthogonally from structures predicted by the AlphaFold2 (AF2)[32] implementation ColabFold[33] and AlphaFold3 (AF3)[35] (where tfmF is mimicked by Tyr in AF2/AF3). Recent assessments of AlphaFold2 showed that > 80% of sidechain conformers predicted with confidence (pLDDT ≥ 70) align with experimental electron density maps[46]. We used the simulated and predicted structures to seek variants possessing short distances between the tfmF label and a proximal aromatic residue with a strong orientational preference. "Positive" designs were defined as computational predictions with at least a 70% preference for a stable perpendicular (shielding ring interaction) or in-plane contact (deshielding ring interaction) and a distance of less

than 6 and 7 Å (5.5 and 6.5 for AF2/AF3 to account for differences between tfmF and Tyr), respectively. Predictions considered "positive" were expected to produce a ring current shift of more than 0.2 ppm (given that most variants without ring currents are dispersed ± 0.2 ppm around the random coil value, Fig. 1c), and were subsequently experimentally assessed.

Our design strategy succeeded in predicting proteins with labelling sites engineered to probe a range of structural motifs. For example, the labelling pair within FLN5 726tfmF 746Phe is positioned across two strands of a β-sheet (Fig. 2b). Consistent with a shielded ring current predicted by both MD force fields (Fig. 2b and Supplementary Tables 5, 6) and both ColabFold/AF3 (Supplementary Tables 3, 4), its experimental $^{19}$F chemical shift is −0.70 ppm relative to random coil (Fig. 2b). Notably, we found that the same tfmF sidechain in β-sheets can even be designed to contact residues on either adjacent strand (Supplementary Fig. 5). Human HRAS labelled with 157tfmF 153His illustrates the ability to design probes within an α-helix (residues $i$ and $i + 4$, Fig. 2c), while we also engineered successful variants to report on protein tertiary contacts between loops/strands and two different α-helices (HRAS 32tfmF 40Tyr and 137tfmF 94His respectively, Supplementary Fig. 6a–f).

### Geometric descriptors of aromatic interactions predict $^{19}$F chemical shifts

To evaluate the overall performance of our design method and the extent to which experimental $^{19}$F chemical shifts can be rationalised solely with ring current effects, we designed and produced a total of 50 different protein variants across six different proteins and one protein complex. These comprised β-sheet proteins FLN5[47], its neighbouring domain FLN4[48], titin I27 (27$^{th}$ Ig-like domain of titin)[49], human filamin A domain 21 (FLNa21)[50], and the FLNa21 complex with human migfilin[50], all α-helical N-terminal domain (NTD) of *E. coli* HemK[51], which folds independently of its C-terminal domain[52], and the mixed α/β human HRAS[53]. For all variants, ColabFold/AF3 predictions and MD simulations with the ff15ipq force field[43] were performed (see "Methods"), and the experimental (secondary) $^{19}$F chemical shifts determined (Supplementary Tables 3–6). Predictions where ring currents were expected, "negative" variants without ring currents, and also tfmF-labels paired with different aromatic residue types were all experimentally tested to accumulate a diverse dataset.

As designed, the variants generally produce $^{19}$F chemical shifts exhibiting a ring current effect at the labelling site whose magnitude correlates with their predicted distances from AlphaFold and MD models (Fig. 2d), and with only minimal changes to their folding stabilities given their high solvent accessibility (Supplementary Table 2). Most of these predictions exhibit a strong preference of tfmF to interact in the perpendicular orientation to the aromatic ring, consistent with nuclear shielding (Fig. 2d). Encouragingly, we find that our design strategy has a high success rate (positive predictive value) of 80, 82 and 90% by ColabFold, AF3 and MD simulations respectively (Fig. 2d, Supplementary Table 7). Thus, the false discovery rate is only 10–20%. The prediction of "negative" variants lacking strong ring currents is less robust, with a false omission rate of 38–45% (55–62% negative predictive value, Fig. 2d and Supplementary Table 7). This is not as crucial for the design strategy, however, as it means some "positive" variants might be missed due to them being predicted as negative variants. When all three prediction methods (ColabFold/AF3/MD) are combined, and a predicted positive design is thus defined as having a ring current by at least one of the prediction tools, and a negative when all three methods predict no ring current, then the false omission rate drops to 26% (Supplementary Table 7). Thus, combined use of the prediction tools may be practical to minimise false omissions. Similar statistics were observed with the C36m MD parameters (Supplementary Fig. 6g, h and Supplementary Table 7). We further note that both true and false ColabFold predictions contain labelling

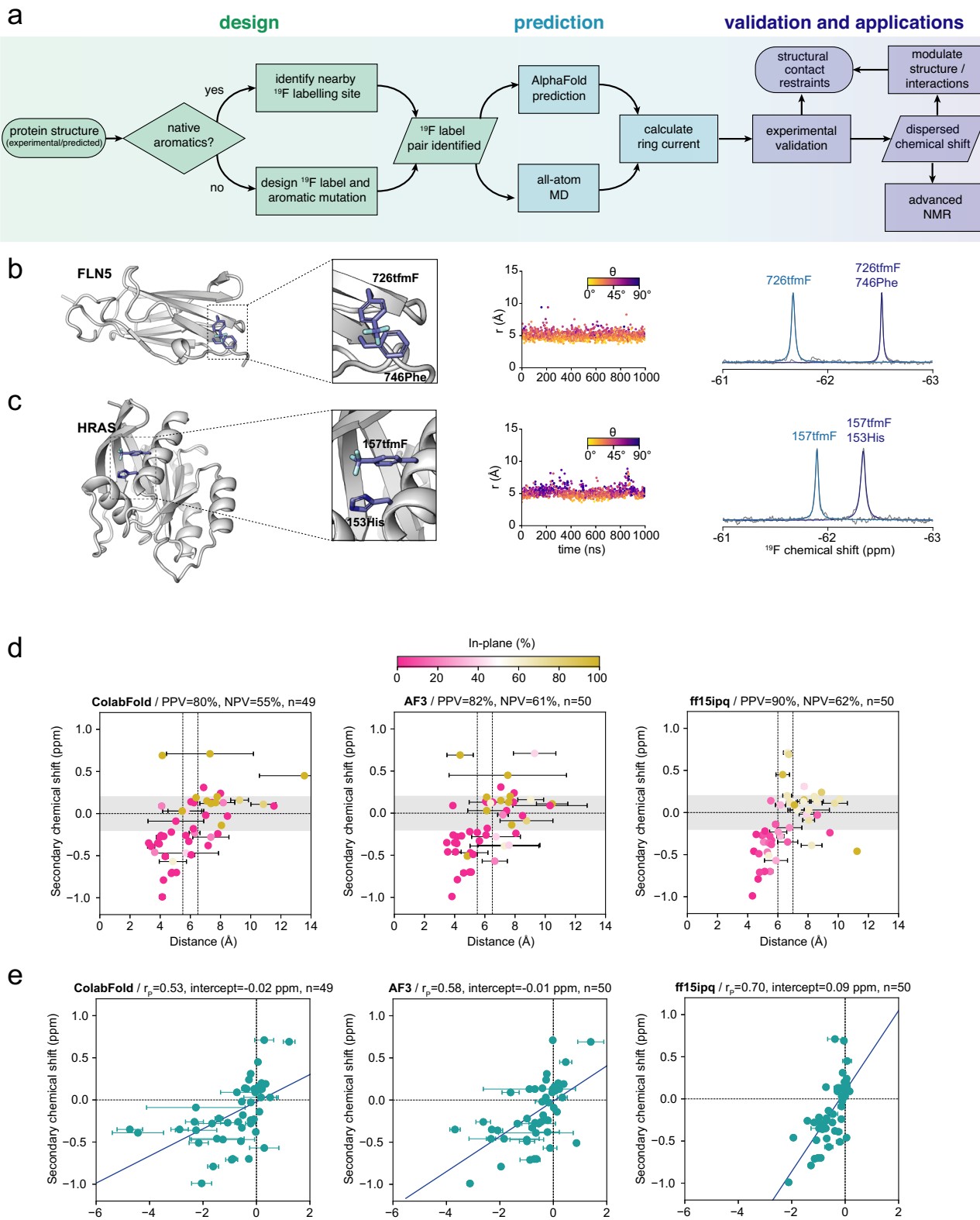

pairs predicted at moderate to high confidence (pLDDT > 70 and > 90, respectively), while AF3 has proportionally fewer moderate-to-high confidence false predictions (Supplementary Table 8), suggesting that AF3-pLDDT values may be useful in prioritising design candidates.

We also evaluated the correlation between the secondary $^{19}F$ chemical shifts and a geometric term quantifying the ring current effect, $(1\text{-}3\cos^2\theta)/r^3$, where $r$ and $\theta$ are the distance and angle of the $CF_3$ group in tfmF relative to the aromatic ring[54], respectively (Figs. 1e, 2e). These analyses show moderately positive correlations (Pearson correlation coefficient, $r_P$) of 0.53 and 0.58 for ColabFold and AF3, respectively. Thus, although AlphaFold predictions perform well in predicting good variants, they are limited in "ranking" variants according to the magnitude of the ring current-induced secondary chemical shift. MD simulations, however, showed a stronger $r_P$ of 0.7,

**Fig. 2 | Rational de novo design of ring current shifts across different structural motifs. a** Flow chart of the design method. **b** (Left) Structural model of FLN5 726tfmF/746Phe, highlighting interaction between 726tfmF and 746Phe across two β-strands. (Middle) Distance ($r$) and angle ($\theta$) between the $CF_3$ group of 726tfmF and aromatic ring of 746Phe observed in a representative all-atom MD simulation. (Right) $^{19}$F NMR spectra of FLN5 726tfmF with and without 746Phe recorded at 298 K and 500 MHz. **c** (Left) Structural model of HRAS 157tfmF/153His, highlighting an interaction between 157tfmF and 153His within an α-helix. (Middle) Distance ($r$) and angle ($\theta$) between the $CF_3$ group of 157tfmF and aromatic ring of 153His observed in a representative all-atom MD simulation. (Right) $^{19}$F NMR spectra of HRAS 157tfmF with and without 153His recorded at 298 K and 500 MHz. **d** Scatter plots (each data point represents a protein variant) correlating the distances (coloured by the fraction of time (MD) spent or models (ColabFold/AF3) in the plane of the ring defined as $\theta > 54.6°$) predicted by ColabFold (left), AF3 (middle) and MD simulations (ff15ipq force field, right) with secondary $^{19}$F chemical shifts for variants of FLN5, FLN4, I27, FLNa21, FLNa21/migfilin complex, HemK, and HRAS. The error bars represent one s.d. over the five predicted models for ColabFold and AF3, and the s.e.m. obtained from three independent simulations for MD. PPV = positive predictive value; NPV = negative predictive value. Positive secondary chemical shift > 0.2 ppm in magnitude. The vertical lines represent the distance cut-off values for perpendicular and in-plane interactions (lower and higher distance, respectively). All variants included solvent-exposed residues on the protein surface. **e** Correlations between predicted geometric factors $(1-3\cos^2\theta)/r^3)$ and secondary $^{19}$F chemical shifts, the corresponding Pearson correlation coefficients ($r_P$) and intercepts for lines of best fit. Errors in all figures represent the s.e.m. from three independent replicates unless otherwise stated.

highlighting that these predictions are more reliable in ranking designed variants (Fig. 2e). More importantly, these results show that ring current descriptors predict secondary $^{19}$F chemical shifts remarkably well (i.e., ~ 50% of the variation in the experimental data is explained by ring currents). The proteins simulated with both force fields (see "Methods") show MD simulations exhibiting correlations of up to 0.8 (64% of variation explained, Supplementary Fig. 6h). The unaccounted variation by our ring current model therefore has likely contributions from other factors such as Van der Waals interactions and electrostatics[28], in line with our observations that removing aromatic rings by mutagenesis reduces the magnitude of the secondary chemical shift by ~75−80% (Figs. 1c, 2b, c). The low RMSD of 0.19 ppm from the regression line observed for the MD (Fig. 2e) is also in line with the ± 0.2 ppm chemical shift range around the random coil value in FLN5 variants without ring currents (Fig. 1c), indicating that other effects such as electrostatics contribute to a negligible extent to the chemical shifts and deviations from the regression line appear to be more related to the accuracy of tfmF-ring interaction predictions. Overall, our approach enables the rational design of proteins with $^{19}$F-labelling sites, whose chemical shifts are engineered to probe distance- and angular-dependent sidechain contacts.

## Applications of the $^{19}$F ring current design strategy

The methodology described above establishes an approach towards rationally designing ring currents and their readout by the resulting $^{19}$F chemical shift of tfmF. Below, we illustrate the approach (and thus the basis for its development) in applications to dynamically interconverting proteins, spanning folding intermediates in both isolated polypeptides and during biosynthesis on the ribosome, ligand- and protein-protein interactions, including within the cellular environment. The harnessing of the method provides valuable structural insights and improves chemical shift dispersions for quantitative studies.

## Discovery of alternative conformational states using ring current design

We first explored whether our ring current design strategy could be employed to elucidate alternative conformational states of proteins, where predictive strategies such as AlphaFold often fail. We examined the *E. coli* N$^5$-glutamine methyltransferase HemK N-terminal domain (NTD), which is composed of a five-helix bundle[51], and considered changes to the local, wild-type structure in response to mutations around helix 3 (h3, I26V/R34K/Q46R). We designed HemK NTD tfmF-labelled at position 38 to report on the structure of h3 by contacting native Phe42 (an $i, i+4$ contact, Fig. 3a, b). The $^{19}$F NMR spectrum of wild-type HemK 38tfmF indeed showed shielding relative to random coil, and the Phe42Ala substitution confirmed the chemical shift is induced by a ring current effect (Fig. 3c). 38tfmF-labelling of the mutant, however, resulted in a random coil $^{19}$F chemical shift, indicating the loss of the local ring current (Fig. 3c), despite $^1$H,$^{15}$N-correlated NMR spectra showing that both variants were globally folded (Supplementary Fig. 7). While ColabFold and AF3 models predict no structural changes (Supplementary Fig. 8), we also performed long-timescale MD simulations of wild-type and mutant HemK (see "Methods"). Six independent simulations, each lasting 20 µs, showed that the protein backbone exhibits increased flexibility in the h3 region of mutant HemK relative to that of wild-type (Fig. 3d), due to local helical unfolding (Fig. 3e and Supplementary 9). The $^{19}$F design, therefore, provides a simple and fast approach to afford local structural insights into conformational changes without the need for multi-dimensional spectral assignment, which may be challenging or infeasible for more complex systems as demonstrated with the following examples.

We next explored the utility of our strategy in the case of a protein folding intermediate of an immunoglobulin-like domain, FLN5. We have previously determined the structural ensemble of a high-energy folding intermediate of FLN5Δ6[37], in which the six C-terminal (G-strand) residues of full-length FLN5 are truncated. This intermediate (I) exhibits increased flexibility (relative to the native state, N) with a disordered C-terminus, while the remaining strands have a native-like conformation (Fig. 3f). We used the 655tfmF labelling site, which experiences a ring current from Phe675 (Fig. 1c–f) to test whether the contacts between the A- and B-strands indeed remain native. The $^{19}$F NMR spectrum shows that the I state has a near-native chemical shift (Fig. 3g), which agrees well with the native-like N-terminal structural model for the I state (Fig. 3f, h).

Having established that ring currents measured by $^{19}$F NMR can be used to characterise partially folded protein conformations with two simple model proteins, we considered an MDa-sized biomolecular complex. Protein folding begins during biosynthesis on the ~ 2.5-MDa ribosome (Fig. 3i), where co-translational folding (coTF) intermediates have been found to be thermodynamically stabilised (relative to off the ribosome) by up to 5 kcal mol$^{-1}$ (refs. 8,9). Such ribosome-bound intermediates have not been directly observable by cryo-electron microscopy due to their dynamic nature, nor by NMR using $^{15}$N or perdeuteration combined with selective $^1$H,$^{13}$C-methyl labelling due to fast transverse relaxation rates induced by even transient ribosome interactions[9,38]. Direct observations have, however, been made using more sensitive $^{19}$F NMR measurements[8], although details of the folding pathway(s) have remained elusive[8,9]. We have previously shown by tfmF-labelling of residue 655, that FLN5 folds co-translationally via two folding intermediates, revealing one intermediate with a random coil chemical shift (I1) and one intermediate with a native-like chemical shift (I2)[8]. These chemical shifts can now be attributed to the absence and presence of the Tyr655-Phe675 contact (Fig. 1c) and, thus, likely an unfolded A-strand and native A-B strand contacts, respectively (Fig. 3j, k).

We explored the I1 and I2 states further using the ring current design strategy and used the engineered ring current arising from 726tfmF/746Phe (described earlier, Fig. 2b), whose large secondary chemical shift directly reports on interactions between the F- and G-strands. By labelling ribosome-bound nascent chain complexes with

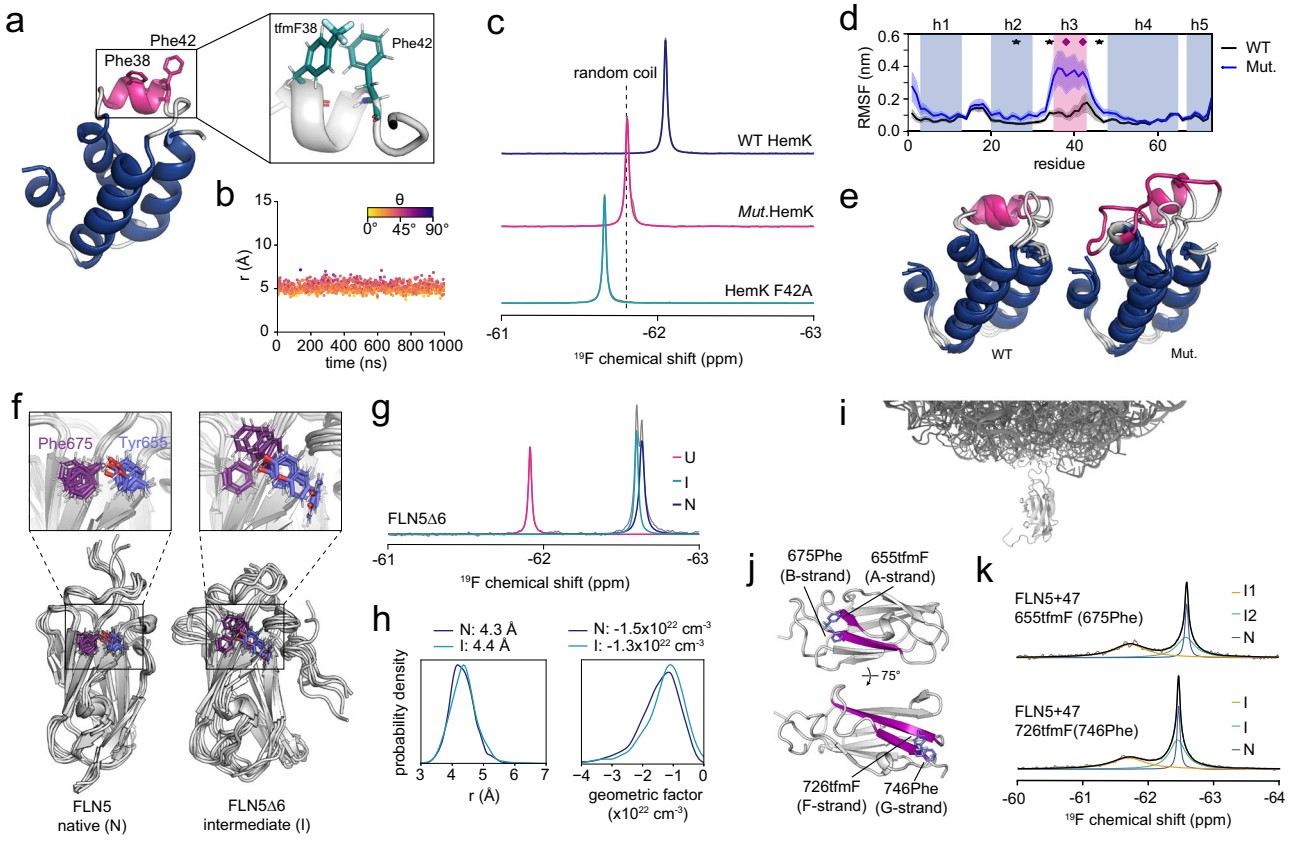

**Fig. 3 | Engineered ring current shifts enable detection and characterisation of alternative protein conformational states. a** Crystal structure (left) of the HemK NTD (residues 1–73, PDB 1T43) and the predicted interaction between 38tfmF and Phe42 observed by MD (right). **b** Distance ($r$) and angle ($\theta$) between the CF$_3$ group of 38tfmF and the aromatic ring of Phe42 observed in a representative all-atom MD simulation. **c** $^{19}$F NMR spectra of HemK NTD 38tfmF variants: *Mut* (harbours I26V/ R34K/Q46R mutations) and the Phe42Ala (below), recorded at 298 K and 500 MHz. **d** Backbone (Cα) root mean square fluctuation (RMSF) analysis for HemK NTD observed in long-timescale all-atom MD simulations (average ± s.e.m. from six independent simulations of 20 µs). The locations of the mutations are indicated by stars, and diamonds represent the tfmF labelling site and aromatic ring location within helix h3. **e** Representative structures of wild-type and mutant HemK obtained from all-atom MD simulations. **f** Structural ensembles of natively folded (N) FLN5 (left) obtained from an all-atom MD simulation using the ff15ipq force field and the FLN5Δ6 folding intermediate (I)[37]. The Tyr655 and Phe675 sidechains are shown as sticks for both ensembles. **g** $^{19}$F NMR spectrum of FLN5Δ6 655tfmF recorded at 283 K and 500 MHz[8,9] showing unfolded (U), intermediate (I) and native (N) conformations at equilibrium. **h** Probability distributions of $r$ and the geometric factor $(1-3\cos^2\theta)/r^3$ calculated for the FLN5 N and I state and average values. **i** Structural model of a folded FLN5 nascent chain tethered to the ribosome. **j** Structural models of FLN5 655tfmF/675Phe and 726tfmF/746Phe highlighting the A/B and F/G strand pairs, respectively. **k** $^{19}$F NMR spectra of FLN5+47 RNCs with the 655tfmF/675Phe and 726tfmF/746Phe labelling pairs, recorded at 298 K and 500 MHz. For FLN5+47 655tfmF/675Phe two folding intermediates (I1 and I2) have been identified previously[8].

this labelling pair, we identified two broad resonances attributable to intermediate states and possessing a random coil and native-like 726tfmF/746Phe chemical shift (Fig. 3j, k). This confirms that one coTF intermediate has an unfolded G-strand, akin to the isolated I state (Fig. 3f). The use of this ring current approach to determine multiple structural restraints and atomistic models of FLN5 co-translational folding intermediates is further explored in ref. 55.

### Detection and structural characterisation of protein-ligand interactions

$^{19}$F NMR is a common tool used in drug discovery and characterising protein-ligand interactions[15]. We tested whether ligand binding to proteins could be detected and structurally interpreted with our method. Noticing that residue 28 of human HRAS points into the ligand (GDP) binding site (Fig. 4a, b), we modelled the structure and dynamics of the HRAS 28tfmF-GDP complex using MD and found that 28tfmF is predicted to be in-range to experience a ring current from the guanine ring system (Fig. 4c and Supplementary Table 5). To consider this experimentally, we purified recombinant HRAS

expressed in *E. coli* and found that HRAS 28tfmF in the presence of GDP/Mg$^{2+}$ exhibits a single peak shielded (− 62.24 ppm) relative to the random coil (Fig. 4d, lower). In the absence of GDP/Mg$^{2+}$ there was an additional peak at − 61.78 ppm corresponding to the apoprotein. The residual amount of GDP-bound HRAS is likely due to the picomolar affinity of GDP for HRAS[56]. Ring currents can thus be used to both directly detect and structurally characterise and validate protein-ligand binding poses, without the need for 2D NMR assignments.

### Structural validation of protein-protein interaction modes in cells

We next investigated the human FLNa21-migfilin complex to explore in-cell protein-protein interactions. Migfilin is a disordered adaptor protein that binds to the FLNa21 filamin, forming an additional strand to one of its β-sheets[50]. The crystal structure of the complex contains two FLNa21 molecules with migfilin sandwiched between the folded domains[50], with FLNa21 chain A suggested by NMR to be the dominant migfilin interaction[50] (Fig. 5a). In order to examine the possible

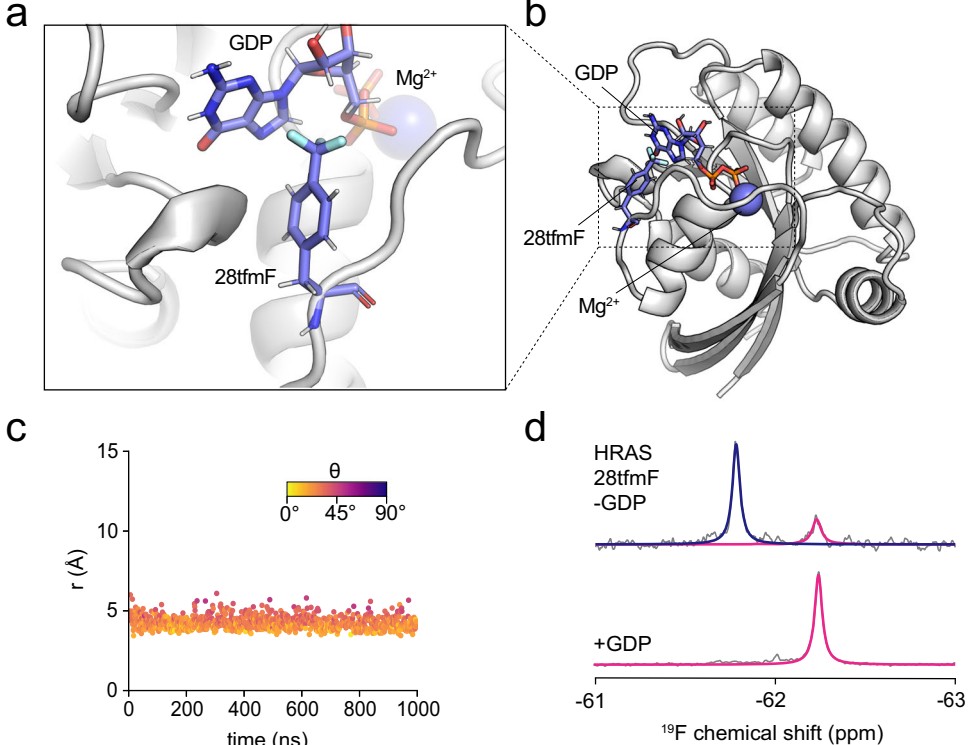

**Fig. 4 | Probing protein-ligand interactions. a, b** Structural model of HRAS 28tfmF bound to GDP obtained from all-atom MD simulations. **c** Distance ($r$) and angle ($\theta$) between the CF$_3$ group of 28tfmF and aromatic rings of GDP observed in a representative all-atom MD simulation. **d** $^{19}$F NMR spectrum of HRAS 28tfmF purified in the absence (top) and presence (bottom) of Mg$^{2+}$/GDP recorded at 298 K and 500 MHz.

interaction mode (including within intact cells), we designed a ring current labelling approach.

Following the outlined workflow (Fig. 2a), an inspection of the crystal structure suggested that the native Phe14 in migfilin could induce a ring current to potential FLNa21 labelling sites on the C-strand (Fig. 5a). MD simulations predicted, however, that regardless of the adjacent labelling site (2274tfmF and 2272tfmF) on the C-strand of FLNa21, no ring current interaction would occur between these sidechains (Fig. 5b, Supplementary Fig. 10 and Supplementary Tables 5, 6). Indeed, titrations of migfilin into FLNa21 2274tfmF in vitro showed only a small $^{19}$F chemical shift difference between the bound and free form of ~0.03 ppm, and which could not be resolved by lineshape analysis (Fig. 5e). Improvement by rational ring current design is therefore necessary, and we screened engineered aromatic variants by MD, predicting that the migfilin Ser12His mutant would produce a strong ring current interaction with 2274tfmF (Fig. 5c, d, Supplementary Fig. 10 and Supplementary Tables 5, 6). This was confirmed experimentally with a titration in vitro, showing the $^{19}$F chemical shift of the bound state is shielded by − 0.49 ppm (Fig. 5e). The increased chemical shift dispersion permits lineshape analysis and accurate quantification of the binding affinity, found to be in the low micromolar range (Supplementary Fig. 11c) in line with previous work[50].

We then sought to use the engineered Ser12His ring current to explore the FLNa21-migfilin interaction in cells. FLNa21 (fused to GST[50]) was co-expressed with migfilin Ser12His, and we clearly observed an additional $^{19}$F NMR peak coinciding with the bound form of the in vitro spectrum (Fig. 5f, Supplementary Fig. 11b). This resonance unambiguously confirms the protein-protein interaction in the crowded intracellular environment and validates the in vitro structural model previously determined[50]. Conversely, co-expression of WT-migfilin did not result in a discerned peak of the bound form in cells (Fig. 5f), due to substantial line broadening paired with a limited chemical shift

difference relative to the free state (0.03 ppm, Fig. 5e). Moreover, 2D $^1$H,$^{15}$N-SOFAST HMQC experiments of in-cell FLNa21 showed no detectable resonances of the domain until after lysis (Supplementary Fig. 11a). The ring current strategy, therefore, enables direct detection and structural interpretation of a protein-protein interaction mode in cells.

## Protein folding thermodynamics determined in cells

We also considered how the ring current strategy might provide useful information about in-cell folding and stability. Cellular environments are known to influence biomolecular interactions, structures, and folding thermodynamics[12,20,21]. The application of in-cell NMR has, however, been limited to a handful of small protein domains with significant line broadening often occurring as proteins interact with other cellular components[12,21,57,58] (Fig. 5). For example, natively folded FLN5 resonances are undetectable in-cell by $^1$H,$^{15}$N-correlated NMR[22], and conversely, intracellular unfolded cross-peaks are severely overlapped using selective $^1$H/$^{13}$C-methyl labelling (Supplementary Fig. 12), both effectively precluding folding equilibrium studies.

Here, we sought to examine how the folding thermodynamics of our model FLN5 protein is altered in vivo using $^{19}$F NMR and thus used the destabilising Phe672Ala mutant to enable a population of both the unfolded and folded states at equilibrium[9]. Despite the broad linewidths of the in-cell spectra (up to ~400 Hz vs 10 Hz observed for purified protein), 655tfmF-labelled FLN5 allowed the resolution (via the induced ring current from Phe675) and the quantification of two conformational (unfolded and folded) populations (Fig. 6a). From these experiments, the equilibrium stabilities and thermodynamic parameters of folding can be extracted in cells (Fig. 6d and Supplementary Fig. 13), and compared to a cell lysate (Fig. 6b) and purified sample (Fig. 6c).

The $^{19}$F experiments show a reduced temperature dependence of in-cell folding and altered stability: FLN5 is destabilised at

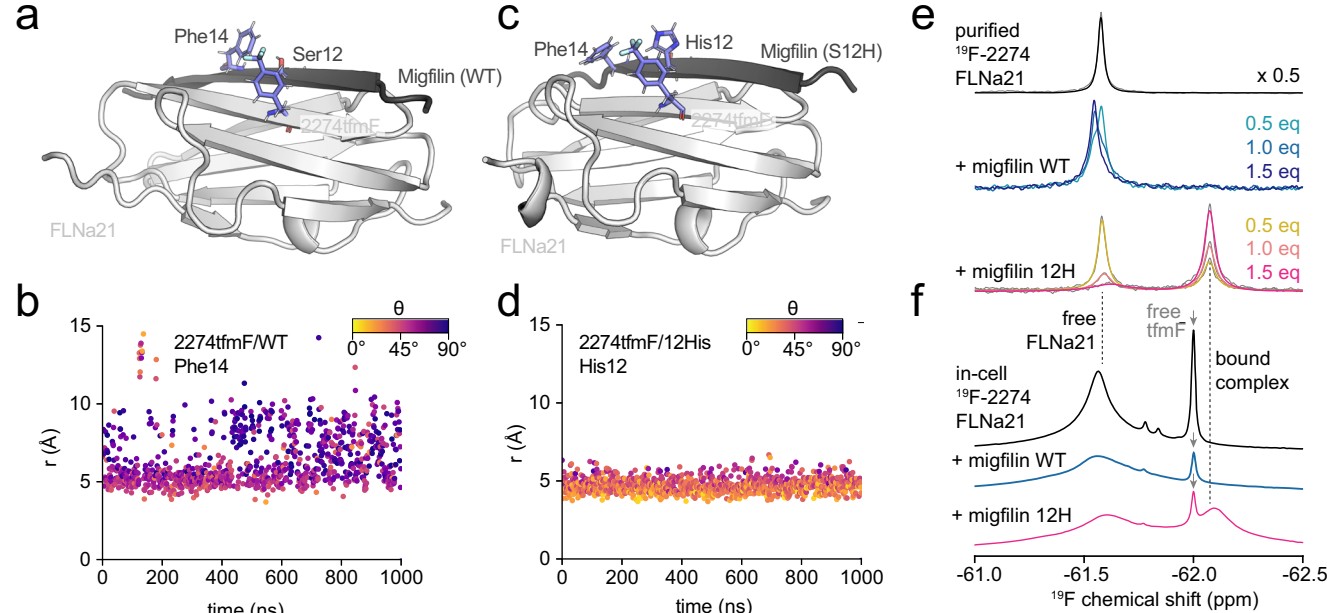

**Fig. 5 | Direct and residue-specific detection of protein-protein interactions in cells. a** Structural model of the FLNa21 (2274tfmF) – migfilin (WT) complex obtained from all-atom MD simulations. **b** Distance ($r$) and angle ($\theta$) between the $CF_3$ group of 2274tfmF and the aromatic ring of Phe14 observed in a representative all-atom MD simulation. **c** Structural model of the FLNa21 (2274tfmF) – migfilin (Ser12His) complex obtained from all-atom MD simulations. **d** Distance ($r$) and angle ($\theta$) between the $CF_3$ group of 2274tfmF and the aromatic ring of 12His observed in a representative all-atom MD simulation. **e** [19]F NMR spectrum of purified FLNa21 2274tfmF alone and with 0.5−1.5 molar equivalents (eq) of migfilin WT and 12His. **f** In-cell [19]F NMR spectrum of FLNa21 2274tfmF and co-expressed with WT and 12H migfilin. Free amino acid (tfmF) is also detected as part of the growth medium. All spectra were recorded at 298 K and 500 MHz.

low temperatures by $+1.4 \pm 0.1\,\text{kcal mol}^{-1}$ (288 K) and stabilised at physiological temperatures (by $-0.4 \pm 0.1\,\text{kcal mol}^{-1}$ at 308 K). Fitting of thermodynamic parameters shows that the enthalpy of folding becomes less negative in cells, and the entropic penalty of folding is reduced (Fig. 6e), an observation that was also found for nascent proteins folding during translation on the ribosome[9]. Thus, in cells, quinary interactions of proteins affect their thermodynamic landscape and energetic factors underlying fundamental biophysical processes such as folding, in line with previous reports of other proteins[12,20,21,59]. These data suggest that further reductions in the entropic penalty of co-translational protein folding[9] occur in the cellular environment, and folding could thus be entropically driven on the ribosome in vivo.

## Discussion

We have presented a rational protein design strategy to yield structurally interpretable [19]F chemical shifts with improved chemical shift dispersions. The approach is relatively straightforward, requiring only the introduction of a commercially available [19]F-label near native or engineered aromatic residues without the implementation of new pulse-sequences nor chemical synthesis. The design can be guided most conveniently and rapidly by AlphaFold at a high success rate (> 80%), with reduced false omissions and higher accuracies achieved by MD simulations, where non-canonical [19]F-labelled amino acids can be explicitly modelled. The method is validated both experimentally, by comparison of tfmF-labelled proteins with and without the accompanying aromatic residue, and by correlating the predicted magnitude of the ring current effect with the observed secondary chemical shifts (Fig. 2). The applications in this work demonstrate a utility in determining how modulations (mutations, ribosome-bound conformations, ligand-unbinding, in-cell environment) affect protein structure and energetics, and future applications may include validation of alternative, predicted structural states, for example, from MD simulations.

Our data illustrate that aromatic ring current effects dominate modulations to the chemical shift of solvent-exposed [19]F NMR labelling sites of cytosolic proteins (Fig. 2d, e), and this is then exploited in our labelling method. The chemical shift of the engineered labels can, therefore, be interpreted as a residue pair-specific Van der Waals contact with an upper bound distance between two amino acids, generally < 7 Å for any ring-current, or a more precise distance determined by MD, where longer simulations and/or enhanced sampling methods[60] may provide improved precision. The information extracted from ring current contacts complements and resembles the interpretation of other types of experimental data probing short-range contacts, such as nuclear Overhauser effects (NOEs), recently developed optical measurements of intramolecular protein distances[61], and longer-range measurements from ([19]F) paramagnetic relaxation enhancement (PRE) NMR, cross-linking mass spectrometry (XL-MS), FRET, and DEER. Upper bound distances of residue pairs, explicit modelling or *post hoc* rotamer calculations may be used to restrain structural models, similar to approaches employed to calculate FRET[62] and PRE/DEER[63] data. Ring current contact data from [19]F NMR could thus be applied as restraints in modelling using MD simulations[64] and directly in AlphaFold-based pipelines as recently illustrated with XL data[65,66]. Further exploitation of our dataset may also enable more detailed chemical shift prediction methods to account for other effects such as electrostatics[28,67].

To ensure general applicability, we have tested our labelling strategy across 6 proteins with different folds. Successful [19]F-ring contacts could be engineered across β-strands, within α-helices and between different secondary structure elements, while the effects of labelling on protein stability were negligible (Supplementary Table 2). Our dataset contains predominantly shielding ring current effects (interactions perpendicular to the ring plane). This may, to some extent, also reflect the intrinsic propensity of tfmF to interact with aromatic rings via π-π stacking interactions for labelling pairs within β-sheets (Fig. 2b) and α-helices (Fig. 2c).

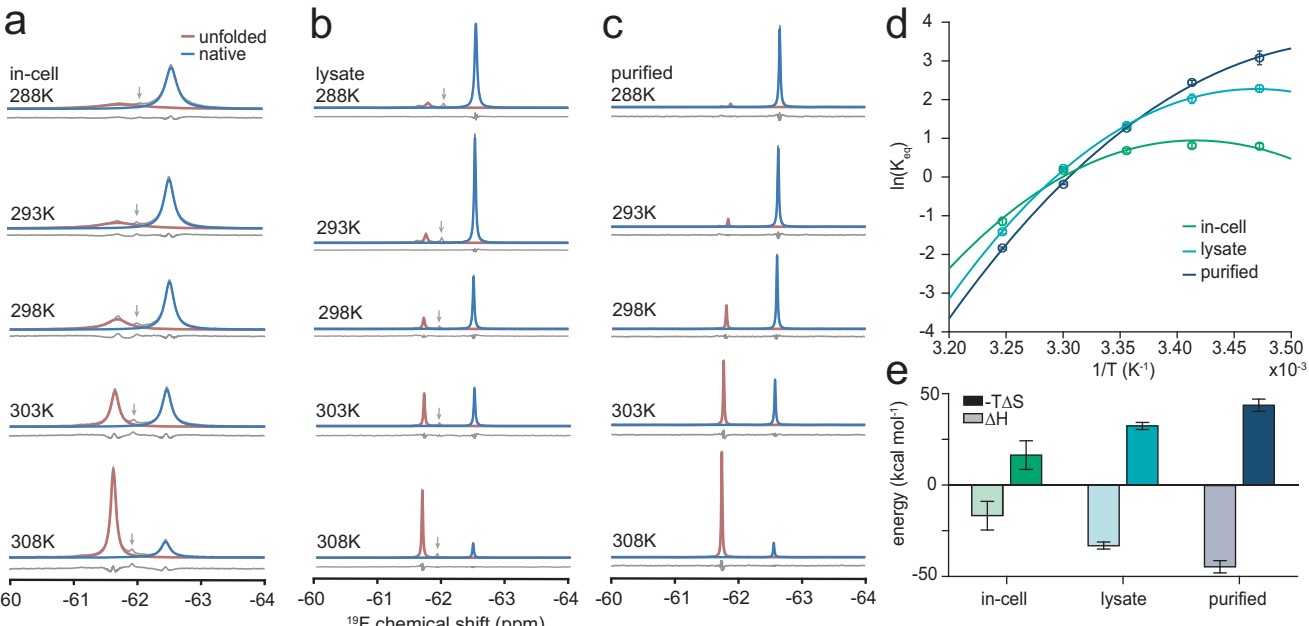

**Fig. 6 | Quantitative protein thermodynamics measured inside living cells. a** In-cell (the arrow highlights free tfmF as part of the growth medium), **b** in-lysate, and (**c**) purified [19]F NMR spectra of FLN5 655tfmF 672Ala[9] at various temperatures recorded at 500 MHz showing an unfolded and folded population. The arrow indicates the position of the free tfmF peak. **d** Temperature-dependence of the folding equilibrium constant ($K_{eq}$) in all conditions. Data were fit to a modified Gibbs-Helmholtz equation (Eq. 1). Data points were calculated from fitted peak integrals from an NMR spectrum of a single biological sample averaged across 256 (purified, lysate) or 512 (in-cell) technical repeats, and errors (standard error) propagated from bootstrapping (200 iterations) of line shape fits. **e** Thermodynamic parameters of protein folding. Values were obtained from fits to the temperature-dependence data in panel (**d**) (from one biological sample), and with errors representing the 95% confidence interval.

The use of tfmF is advantageous due to its low number of rotatable sidechain dihedral angles, and our strategy is likely to be extended to other, similar fluorinated amino acids in the future, such as recently developed pyridone-based[17] and monofluorinated[3] probes. The approach is independent of the incorporation methods, whether genetically encoded, by selective pressure incorporation or by covalent conjugation. While we have focused on phenylalanine and histidine as engineered aromatic rings due their smaller sizes compared to tyrosine and tryptophan (and the slightly stronger ring current intensity of Phe compared to Tyr[68]), exemplar variants with natural aromatic sidechains show these residues do not preclude induction of ring currents (e.g., native Trp2262 in FLNa21 tfmF2242 and Tyr40 in HRAS 32tfmF, Supplementary Table 5).

We have showcased the capabilities of our method by examining a range of different biological systems. Our exemplar applications included the identification of alternative conformational states, the characterisation of nascent protein folding intermediate structures on the ribosome, and probing protein-ligand and protein-protein interactions; each of which have permitted structural characterisation or validation, and where [19]F NMR is the only available experimental method capable of directly obtaining residue-resolution data of co-translational folding intermediates outside the ribosome exit tunnel[8,9]. Our method also overcomes challenges with spectral overlap associated with $CF_3$-probes, and this has permitted in-cell NMR measurements to validate in vitro structural models and quantify thermodynamics in the cell.

Rationally designed fluorinated proteins together with simple 1D pulse-acquire experiments permit access to structural characterisations of complex biological systems, where multi-dimensional [19]F NMR methods can suffer from sensitivity limitations[1,31]. We anticipate future applications of more advanced NMR experiments will include, amongst other measurements, those of protein dynamics across different time scales; for example, CEST experiments[8] will particularly benefit from the increased spectral dispersion of chemical shifts of different conformational states. The rational design of [19]F-labelling sites and our chemical shift dataset will, therefore, support technical developments and biological applications of NMR spectroscopy.

## Methods
### Molecular biology and protein production
All mutations (including amber stop codons) were introduced using standard site-directed mutagenesis procedures, and constructs were confirmed by DNA sequencing. FLN5, FLN4, titin I27, HemK NTD (residues 1–73), and HRAS were expressed and purified as hexahistidine-tagged constructs as previously described[8,9,38]. Expression of GST-tagged FLNa21[50] was performed as for the other proteins and purified using a similar procedure but replacing the Ni[2+]-NTA resin with a Glutathione Sepharose™ 4B resin (Cytiva™). Elution was achieved in the presence of 10 mM reduced L-glutathione. Migfilin (residues 1–85) DNA constructs and production were previously described[50]. Labelling with 4-trifluoromethyl-L-phenylalanine (tfmF) was achieved by amber suppression, as previously described[8], where protein expressions were modified as follows: BL21(DE3) *E. coli* cells were co-transformed with the pEVOL-pCNF-RS suppressor plasmid[69,70], and expression performed in LB media (for [19]F-labelling only) or M9 media (for additional uniform [15]N-labelling). Expression of the orthogonal pair was immediately induced by the addition of L-arabinose (0.2% (w/v)) at the start of the culture growth and incubated until OD600 ~ 0.5 before the addition of tfmF (0.5 mM), and a further incubation until OD600 ~ 0.6 when expression of the protein of interest was induced by addition of IPTG (1 mM). The same expression conditions were used as for unlabelled samples, typically 4 h at 37 °C. Yields for [19]F-labelled proteins were typically 50-100% of those for unlabelled proteins; specifically, ~ 5 mg/L for FLN5, ~ 5 mg/L for FLN4, ~ 2 mg/L for FLNa21-GST, ~ 1 mg/L for titin I27, ~ 1 mg/L for HemK, ~ 2 mg/L for HRAS, and ~ 0.3 mg/L for migfillin.

**Table 1 | List of proteins simulated with the C36m force field including their PDB/AlphaFold2 database codes and references**

| Protein (residues) | Structure code | Source | Reference |
|---|---|---|---|
| FLN5 (650–660) peptide | N/A | Linear peptide built in PyMol | N/A |
| FLN5 (645–750) | 1QFH/6G4A | X-ray crystallography/NMR | 37,47 |
| FLN4 (547–648) | AF-P13466-F1 | AlphaFold2 | 32,97 |
| FLNa21 (2236–2328) | 2WOP | X-ray crystallography | 50 |
| FLNa21 (2236–2328) / Migfilin (8–16) complex | 2WOP | X-ray crystallography | 50 |
| I27 (1–89) | 1TIT | NMR | 49 |

The FLN5 (650–660) peptide was used as a reference to calculate the solvent accessibility of the CF$_3$ group in tfmF for an unfolded peptide.

## NMR spectroscopy

NMR data were acquired on 500- and 800-MHz Bruker Avance III spectrometers, both equipped with TCI cryoprobes and all recorded using TopSpin3.5pl2 at 298 K unless stated otherwise. All samples were measured in Tico buffer (10 mM HEPES, 30 mM NH$_4$Cl, 12 mM MgCl$_2$, 1 mM EDTA, 2 mM β-merceptoethanol, pH 7.5) supplemented with 10% (v/v) D$_2$O and 0.001% (w/v) sodium trimethylsilylpropanesulfonate (DSS) as the reference compound. 1D $^{19}$F pulse-acquire experiments were recorded with an acquisition time of 350 ms and a recycle delay of 1.5 s. To monitor the stability of the FLN5+47 726tfmF/746Phe RNC, short successive spectra were recorded, such that data with detectable changes over time (from nascent chain release or degradation) were discarded, and only data from intact RNC complexes were summed together, as previously described[8,9]. 2D $^1$H, $^{15}$N SOFAST-HMQC experiments were performed with direct and indirect dimension acquisition times of 50.4 and 295 ms, respectively, and a recycle delay of 100 ms. Data processing and analysis were performed with nmrPipe[71], CCPN Analysis[72], MATLAB (R2014b, The MathWorks Inc.) and Julia[73] as previously described[8]. The time-domain $^{19}$F NMR data were multiplied with an exponential window function with a line broadening factor of 5−10 Hz prior to Fourier transformation, and subsequently, baseline corrected, and line shapes analysed using Lorentzian functions[8]. Resonances were assigned based on our previously deposited resonance assignments, recorded under identical conditions: FLN5 amide backbone (BMRB34249) and FLN5 side chains (BMRB51075).

## In-cell NMR spectroscopy

High-density *E. coli* cell cultures for in-cell NMR were prepared using the protocol for the production of purified RNC samples as previously described[8,9,38,74]. Protein expression was induced with 1 mM IPTG (0.25 mM for FLNa21+migfilin cell cultures where both proteins were induced with IPTG) for ~16 h at 30 °C. Cells were then harvested by centrifugation at 4000 rpm for 15 min, washed three times with and subsequently resuspended in in-cell NMR buffer (75 mM bis-tris propane, 75 mM HEPES, 25 mM MgCl$_2$, pH 7.5). Cells slurries were prepared at a concentration of 40% (w/v) and supplemented with 10% D$_2$O and 0.001% (v/v) DSS. To monitor sample stability, we recorded short spectra in succession to identify spectral changes over time; no discernible changes were observed between spectra acquired during the course of NMR data acquisition. Cell leakage of $^{15}$N-labelled samples was monitored using 1D $^1$H,$^{15}$N-correlated SORDID diffusion measurements interleaved between 2D SOFAST-HMQC experiments and acquired with a diffusion delay of 300 ms, using gradient pulses of 4 ms and gradient strengths of 5% and 95% of the maximum gradient strength (0.56 Tm$^{-1}$)[22,75]. Given the sensitivity limitations of $^{19}$F diffusion measurements, we instead assessed cell leakage of $^{19}$F-labelled samples by recording equivalent NMR experiments of the supernatant after centrifugation at 9000 rpm for 10 min. Acquisition parameters for all experiments were identical to those performed on purified samples described above. For the quantification of protein folding thermodynamics, we used MATLAB to fit the temperature-dependent folding equilibrium constant, $K_{eq}$ (directly measured from peak integrals), to a modified Gibbs-Helmholtz equation to determine the change in entropy ($\Delta S$), enthalpy ($\Delta H$) and heat capacity of folding ($\Delta C_p$) of folding:

$$\ln\left(K_{eq,T}\right) = -\left(\frac{\Delta H_{T0} + \Delta C_p(T - T0)}{R}\right)\left(\frac{1}{T}\right) + \left(\frac{\Delta S_{T0} + \Delta C_p\ln(\frac{T}{T0})}{R}\right)$$

(1)

$T$ and $T_O$ are the temperature of the measurements and standard temperature (298 K), respectively. We report the errors of the fitted parameters as the 95% confidence interval (CI).

## MD simulations of fluorinated protein variants

All MD simulations were performed using GROMACS version 2021 or 2023[76]. Independent simulation replicates were initiated from different sidechain rotamers of either the fluorinated amino acid and/or nearby aromatic residue where possible. Proteins listed in Table 1 (and their fluorinated variants) were parametrised using the CHARMM36m (C36m) force field[44], including parameters for 4-trifluoromethyl-L-phenylalanine (tfmF, July 2021 release of the C36m force field files for GROMACS)[45]. Simulations were initiated from the following experimental structures, after introducing tfmF and the relevant mutations in silico using PyMol (version 2.3, Schrödinger, LLC). The C36m force field resulted in the partial unfolding of wild-type HemK and HRAS, which were, therefore, not further simulated with C36m. We thus describe the analysis of all proteins (with ff15ipq, see below) in the main text and a common dataset comparison with C36m in the Supplementary Information.

Protein molecules were placed in the centre of a dodecahedral-shaped box at least 1.2 nm away from the box edge. The systems were then solvated, and MgCl$_2$ was added at a final concentration of 12 mM to neutralise the simulation box. We note, however, that the choice of salt should not be a critical parameter for ring current predictions since interactions between neutral (aromatic) sidechains are analysed (and consistent results are observed with the ff15ipq force field/setup described below, Supplementary Fig. 6g, h). Energy minimisation was run using the steepest descent algorithm. Dynamics simulations then employed the leap-frog integrator and a 2 fs timestep along with the LINCS algorithm[77] to constrain all bonds connected to hydrogen. Non-bonded interactions were cut off at 1.2 nm, and an additional switching function between 1.0 and 1.2 nm was applied to Lennard-Jones interactions. The Particle Mesh Ewald (PME)[78] method was used for long-range Coulombic interactions with cubic interpolation and a fourierspacing of 0.16 nm. The temperature was kept constant using the velocity rescaling algorithm[79] with a time constant of 0.1 ps. The simulation systems were then first equilibrated in the NVT ensemble for 500 ps at 298 K using position restraints on all heavy atoms (1000 kJ mol$^{-1}$ nm$^{-2}$ along all coordinate axes), followed by an additional 500 ps in the NPT ensemble at the same temperature and with the same restraints. The pressure was controlled using the Berendsen barostat[80], set to 1 bar with a compressibility of $4.5 \times 10^{-5}$ bar$^{-1}$ and coupling constant of 2 ps. Production simulations in the absence of any restraints were then run for 1 μs in the

**Table 2 | List of proteins simulated with the ff15ipq force field including their PDB/AlphaFold2 database codes and references**

| Protein (residues) | Structure code | Source | Reference |
|---|---|---|---|
| FLN5 (645–750) | 1QFH/6G4A | X-ray crystallography/NMR | 37,47 |
| HemK (1–73) | 1T43 | X-ray crystallography | 51 |
| FLN4 (547–648) | AF-P13466-F1 | AlphaFold2 | 32,97 |
| FLNa21 (2236–2328) | 2WOP | X-ray crystallography | 50 |
| FLNa21 (2236–2328) / Migfilin (8–16) complex | 2WOP | X-ray crystallography | 50 |
| I27 (1–89) | 1TIT | NMR | 49 |
| HRAS (1–166, Mg²⁺-GDP) | 4Q21 | X-ray crystallography | 53 |

NPT ensemble using the Parrinello-Rahman algorithm[81], saving protein coordinates every 0.1 ns.

The following proteins in Table 2 were simulated using the AMBER ff15ipq force field[42] and SPC/$E_b$ water parameters[82]. Parameters for tfmF were taken from ref. 43. Systems were prepared using *tleap* from AmberTools23[83]. Proteins were placed in a truncated octahedral box and neutralised with sodium ions. The ParmEd python library[84] was then used to convert the AMBER files to GROMACS format. Simulations were conducted using a 1.0 nm cut-off distance for nonbonded interactions and the PME method[78] for long-range Coulombic interactions with cubic interpolation and a fourierspacing of 0.125 nm. The remaining run parameters and protocols were identical to the C36m simulations described above. As mentioned above, HemK and HRAS (in complex with Mg²⁺-GDP) were only simulated with ff15ipq parameters as we observed protein instability during microsecond timescale MD simulations with C36m for these systems.

**GDP parameterisation**

For the HRAS complex, GDP bonded, and Lennard-Jones parameters were assigned based on the GAFF2 force field[85] with *antechamber*[86]. Using AM1-BCC partial charges[87] as a starting point, we then parameterised implicitly polarised charges (IPolQ)[88] to obtain GDP charges that are compatible with the ff15ipq force field[42]. A protocol described in the following GitHub repository was followed: https://github.com/darianyang/ff15ipq-lig. AMBER20[89] and ORCA5.0.4[90] were used for parameterisation and quantum mechanical (QM) calculations, respectively. Briefly, 40 GDP conformers were sampled in a 20 ns unrestrained NPT simulation at 450 K and 1 bar with the SPC/$E_b$ water model[82]. The electrostatic potential in vacuum and explicit solvent (surrounding the solute within 5 Å) was then calculated for each conformation as previously described[43], and partial charges fit to the electrostatic potentials with the mdgx programme[89] while restraining the total charge to −3.0 and enforcing equal charges for chemically equivalent atoms. The vacuum and solvated partial charges were then averaged to obtain the IPolQ charges. The protocol was repeated with the resulting charges until the charges converge (<10% change with respect to the previous iteration). The final partial charges used in this work are listed in Supplementary Table 1. We verified that these GDP parameters resulted in a stable HRAS-GDP complex for wild-type HRAS, including proper coordination of the bound Mg²⁺ ion, in agreement with the X-ray structure[53]. This was observed over microsecond-long simulations that are required for assessing fluorinated HRAS variants (Supplementary Fig. 4).

**Long-timescale MD simulations of HemK**

Starting from the crystal structure (PDB 1T43, residues 1–73), we centred the N-terminal domain (NTD) of HemK in a dodecahedral-shaped box at

least 1.2 nm distanced from the edge of the box. Water molecules and MgCl₂ (12 mM) were then added to solvate and neutralise the system. The protein was parameterised with the DES-Amber force field[91] and TIP4P-D water model[92]. Energy minimisation was performed using the steepest descent. In the following MD runs, the leap-frog integrator and a 2 fs timestep, along with the LINCS algorithm[77] to constrain all bonds connected to hydrogen, were used. Nonbonded Van der Waals interactions were treated with a 0.9 nm cut-off. Electrostatics were calculated using a 1.0 nm cut-off for the real-space contribution, and the Particle Mesh Ewald (PME)[78] method was used for long-range interactions with cubic interpolation and a fourierspacing of 0.125 nm. The temperature/pressure coupling parameters and equilibration protocol were as described above for the fluorinated protein variants. Six independent production simulations were launched starting with different initial velocities at the equilibration stage for 20 μs per replicate in the NPT ensemble (totalling 120 μs of sampling per variant). For the production runs, we employed a 4 fs timestep with hydrogen mass repartitioning applied[93]. Protein coordinates were saved and analysed every 1 ns, resulting in 20,000 snapshots per trajectory.

**Sidechain modelling of BTFMA**

A model structure of a cysteine conjugated to BTFMA was first built in PyMol (version 2.3, Schrödinger, LLC) by mutating residue 655 in FLN5 (PDB1QFH) to cysteine and completing the sidechain. The amino and carboxylate terminal groups were capped with acetyl and N-methyl amide groups, respectively. ACPYPE was used to assign GAFF2 force field parameters[86,94]. Using the same nonbonded interaction cut-off values and equilibration protocols as for our AMBER-based MD simulations detailed above, we performed a 100 ns production simulation with position restraints (1000 kJ mol⁻¹ nm⁻² along all coordinate axes) applied to the N, CA and C atoms to sample different sidechain conformations. Cys-BTFMA atoms were saved every 100 ps resulting in 1000 snapshots of sidechain conformations. These were aligned to backbone atoms (N, CA, C) of residue 655 in FLN5, and conformations that resulted in clashes (< 2.5 Å for heavy atom distances) were discarded. Using these aligned rotamers we calculated the distribution of distances between the CF₃ group of Cys-BTFMA and the Phe675 aromatic ring.

**AlphaFold structure predictions**

We used the AlphaFold2[32] and AlphaFold-Multimer[34] implementation ColabFold v1.5.5[33] to predict structures of all fluorinated protein variants and complexes by substituting tfmF with a tyrosine residue. All predictions were run on the Google Colab platform (https://colab.research.google.com/github/sokrypton/ColabFold/blob/main/batch/AlphaFold2_batch.ipynb) using T4 GPUs and applying default settings with templates. All five models were relaxed for each variant. We also used AlphaFold3[35] (AlphaFold-beta-20231127 version, https://alphafoldserver.com/) for all proteins and complexes. For HRAS, we explicitly included the ligands (Mg²⁺ and GDP). Default settings were applied.

**Analyses of MD simulations and predicted structures**

We calculated the Cα-RMSD with respect to the energy-minimised input structure for all simulations to ensure that the proteins remained stable and folded throughout the MD runs using the *gmx rms* tool[76]. The solvent-accessible surface area (SASA) for the CF₃ group of tfmF was computed using the *gmx sasa* tool[76]. The root mean squared fluctuations (RMSF) of the protein backbone (Cα atoms) were computed using the gmx rmsf tool. The distances and angular orientation of the CF₃ group (or the tyrosine OH atom for AlphaFold predictions) relative to nearby aromatic sidechains were calculated using in-house Python scripts and the MDAnalysis package[95,96] (see Code Availability). We used the centre of mass of the CF₃ group (or the tyrosine OH atom) and the centre of mass of the neighbouring aromatic ring heavy

atoms for these calculations (including for two-ring systems like Trp and GDP to approximate the average position relative to the two 5- and 6-membered rings). Additionally, we calculated a geometric factor quantifying both the distance and angular contributions to the expected ring current effects within one term[54], $(1-3\cos^2\theta)/r^3$, to more directly assess the correlation between the experimentally measured chemical shifts and computational predictions. Distances, angles and SASA values were calculated for every 1 ns in the trajectories. We calculated averages and standard errors from three independent simulation replicates, discarding the first 200 ns of each replicate to allow for equilibration of the sidechain rotamers. Positive variants were defined as variants that exhibited a secondary chemical shift of at least 0.2 ppm in magnitude. From the computational side, we defined positive predictions to have a distance of no more than 6 and 7 Å for perpendicular and in-plane interactions, respectively, with an orientational preference of more than 70% (i.e., at least 70% in-plane or 70% perpendicular). The distance cut-offs for AF2/AF3 predictions were 5.5 and 6.5 Å, respectively, to approximately account for the difference in non-clashing distances between a $CF_3$ and OH group. Performance statistics were calculated based on these classifications.

### Reporting summary

Further information on research design is available in the Nature Portfolio Reporting Summary linked to this article.

## Data availability

All [19]F NMR chemical shift data are available in the Supplementary Information (Supplementary Tables 3–6). AlphaFold-generated structural models, their confidence scores, and MD trajectories are available on Zenodo at https://zenodo.org/records/14288915. Distance and angular data calculated from the structural models and MD simulations (e.g., Fig. 1f) are also included in the Zenodo repository. The Zenodo repository contains all ColabFold and AlphaFold3 predictions in the 'AlphaFold' directory, where for each variant, a subdirectory contains the structures and a CSV file summarising distances and angles describing the interactions between the tfmF/aromatic pair. For all MD trajectories ('MD_C36m' and 'MD_ff15ipq' directories), the distances and angles are included as NPY files in each subdirectory. Additional source data (Figs. 1c, 3d, 6d, e) are provided as a Source Data file. This study made use of the following public datasets deposited in the protein databank (PDB, https://www.rcsb.org/): 1TIT (titin I27), 2W0P (FLNa21-migfilin complex), 1QFH (FLN5 crystal structure), 6G4A (FLN5 NMR structure), 1T43 (HemK NTD), 4Q21 (HRAS). The study also made use of the following NMR assignments deposited in the Biological Magnetic Resonance Data Bank (BMRB, https://bmrb.io): BMRB34249 (FLN5 backbone), BMRB51075 (FLN5 sidechains). Source data are provided in this paper.

## Code availability

A Python script to calculate distances, angles, and geometric factors for ring current predictions is available on GitHub (https://github.com/julian-streit/RingCurrents19F) and Zenodo (https://doi.org/10.5281/zenodo.15173797). MATLAB scripts to process and fit [19]F NMR spectra are also available on GitHub (https://github.com/shschan/NMR-fit) and Zenodo (https://doi.org/10.5281/zenodo.15169089).

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

## Acknowledgements

This study was funded by a Wellcome Trust Investigator Award (to J.C., 206409/Z/17/Z). We acknowledge the use of the UCL Biomolecular NMR Centre. Computational resources were provided by the Baskerville Tier 2 HPC service (https://www.baskerville.ac.uk/). Baskerville was funded by the EPSRC and UKRI through the World Class Labs scheme (EP/T022221/1) and the Digital Research Infrastructure programme (EP/W032244/1) and is operated by Advanced Research Computing at the University of Birmingham. We are also grateful to the UK Materials and Molecular Modelling Hub for computational resources, which is partially funded by the EPSRC (EP/T022213/1, EP/W032260/1 and EP/P020194/1), and the UCL Kathleen High Performance Computing Facility (Kathleen@UCL), and associated support services. We thank Prof. M. Rodnina (Max Planck Institute for Biophysical Chemistry, Göttingen) and Prof. D. Calderwood (Yale School of Medicine) for the kind gift of the HemK and human migfilin plasmids, respectively.

## Author contributions

J.O.S., S.H.S.C., and J.C. designed the project. J.O.S., S.H.S.C., and S.D. produced the protein samples and performed NMR experiments. J.O.S. performed and analysed the MD simulations and AlphaFold predictions. S.H.S.C. analysed the NMR experiments. J.O.S., S.H.S.C., and J.C. sourced the computational resources and funding. S.H.S.C. and J.C. supervised the project. J.O.S., S.H.S.C., and J.C. prepared the manuscript with input from S.D.

## Competing interests

The authors declare no competing interests.
