## [Transparent Peer Review file · Nature Communications]

Rational design of ^{19}F NMR labelling sites to probe protein structure and interactions

Corresponding Author: Professor John Christodoulou

Version 0:

Reviewer comments:

Reviewer #1

(Remarks to the Author)

The article entitled "Rational design of ^{19}F NMR labelling sites to probe protein structure and interactions" by Julian et al. have systematically explored the strategic utilization of ring current effect of the aromatic residues to enhance the chemical shift dispersion of ^{19}F labels. ^{19}F NMR has been shown to have unique advantages to study complex protein targets, including multiple conformational dynamics in membrane proteins, protein folding of nascent chains on ribosome, protein-protein interaction in cells, etc. However, a significant limitation of highly sensitive and widely used $-\text{CF}_3$ based labels is the relatively small chemical shift dispersion, i.e., the extent of the response of its chemical shift to the environments, resulting in severe peak overlapping or inability to resolve the environment changes. Although the ring current effect has been found to be an important factor to modulate the ^{19}F chemical shift, it has never been systematically explored, and a reliable method to efficiently find the optimal ^{19}F labeling sites to monitor the protein behavior is still missing. To overcome this limitation, Julian et al. leverage the strength of AlphaFold and all-atom molecular dynamics to obtain a generalized geometric factor between the ^{19}F label and strategically chosen intrinsic or engineered aromatic residues and use the geometric factor to evaluate the potential secondary ^{19}F chemical shift changes. The method achieved an excellent success rate, and the authors have proved its power on several challenging applications. Particularly, the observation of the interaction between FLNa21 and migfilin and the folding thermodynamics of FLN5 in living cells is amazing. I believe that this method will greatly benefit the broad community of using ^{19}F NMR to study protein dynamics and protein-protein/ligand interactions. The manuscript is well prepared, however, I do have a few questions.

1. While the comparison between Phe and His clearly showed the larger chemical shift changes, it would be great if the authors could use one representative case to compare Try over Phe, although Try is generally considered as less aromatic than Phe.
2. In line 119, the chemical shifts are restricted only to solvent-exposed labeling sites. Are the analyses in Figure 2 restricted to solvent-exposed sites too? If it's true, the authors may need to give some explanations.
3. Figure 2e clearly shows correlation between the secondary ^{19}F chemical shifts and the geometric factor. For those with large deviations from the regression line, could they be rationalized by some significant factors like charge, polarity, or other effects?
4. As an important method, some more details of experimental description would be very useful for readers and potential users. Especially considering so many expressions have been done for the incorporation of tmF for this manuscript, could you mention in the Methods section a little about the protein yields?

Minor issue:

In Figure 6d, the colors of in-cell and lysate are indistinguishable, at least on my screen.

(Remarks on code availability)

I am not good at coding, so it's out of my ability to access the quality of the code.

Reviewer #2

(Remarks to the Author)

In this work, it is shown that ring currents affect the ^{19}F chemical shift of trifluoromethyl phenylalanine, making it a sensitive probe of protein conformation. The authors exploit this property and establish predictive computational tools that select artificial protein mutants in which the chemical shift of tmPhe is highly sensitive to small changes in local protein conformational states. This approach is then applied to investigate several different phenomena, including the identification of alternative conformational states, the characterisation of nascent protein folding intermediate structures on the ribosome, and probing protein-ligand and protein-protein interactions both in vitro and in bacterial cells. This is an important advancement that will greatly extend the applicability of ^{19}F NMR spectroscopy to investigate protein conformation and interactions, especially in challenging conditions such as with large complexes or in the cellular environment. The work is elegantly designed, properly executed and clearly written, and the conclusions are fully supported by the experimental data.

Therefore, I recommend publication in Nature Communications, after few minor issues are addressed, mainly to improve the clarity and readability of text and figures.

Minor points:

1) Although very pleasing to the eye, the choice of colours in some figures makes them hard to interpret: in Figure 1 panel c (top), shades of indigo-blue-turquoise are used to represent eighteen overlaid spectra. It is impossible to tell them apart even for a non-colourblind person. A similar problem is found in Figure 6d. Similarly, in Figure 1 f the colour coding chosen to represent the angle makes it hard to understand how variable the angle is in the MD simulation: angles between 0 and 45° almost look the same colour, even with a properly calibrated screen. The same issue applies to the same type of plot in all subsequent figures.

2) In all distance and angle plots of representative all-atoms MD simulations shown throughout the work, it would be useful to report the average distance and angle values calculated from the MD simulation, as they constitute the selection criterium for the mutations (if I understood correctly).

3) "...from structures predicted by AlphaFold2 (AF2)/ColabFold..." How does ColabFold differ from AlphaFold2? As I understand, the former is an accessible implementation of the latter, so maybe this could be rephrased to make it clearer.

4) I had trouble understanding the construction of this sentence: "We further note that while labelling/residue pairs that are predicted with ColabFold at moderate to high confidence (pLDDT > 70 and > 90, respectively) for both true and false predictions, AF3 has proportionally fewer moderate-to-high confidence false predictions" Is a verb missing in the first half?

5) In Figure 3c, the dashed line corresponding to the random coil chemical shift is almost invisible.

6) Figure 3f might benefit from insets zooming in on the conformations of Phe and Tyr residues, which are otherwise drawn too small to be understood.

7) In Figure 4c, as GDP has two aromatic rings, with respect to which axis was the angle calculated?

8) In Figure 5 a and b, I cannot tell which chains belong to FLNa21 and which to migfillin. They have too similar shades of grey. Also, aromatic residues could be coloured consistent with the previous figures.

(Remarks on code availability)

Reviewer #3

(Remarks to the Author)

In the current work, Streit and coworkers use computational modeling to guide their labeling strategy for their experimental ^{19}F protein NMR methodology. They demonstrate high success rates of predicted ring current-induced chemical shift (CS) changes (relative to random coil values) using either AlphaFold models or MD simulations, with the latter being even more accurate in predicting CS changes. The demonstrated combined computational/experimental approach improves the CS dispersion and enables simple 1D NMR experiments to analyse structural features of alternative protein conformational states. This is shown with impressive examples ranging from ribosome-bound folding intermediates, and even in-cell NMR to probe protein interactions and folding thermodynamics.

The authors followed established state-of-the-art procedures to set up, run, and analyse their MD simulations. It is convincing that the authors used two current biomolecular force fields that are suitable for simulating also fluorinated amino acids, ff15ipq and C36m, which in most cases yielded consistent results. The simulation times of 1 microsecond per replicate (or even (much) longer for HemK, 20 microseconds per replicate) are state-of-the-art as well. Importantly, the authors also ran independent replicate simulations initiated from different rotamer states of the sidechains of interest, which (again) yielded consistent results. This shows the convergence of the simulations, and altogether it underlines the robustness of the simulation results. At the end of the day, most convincing is of course the experimental validation (or verification) of the predictions from the simulations. They can thus contribute to a rational design of strategic ^{19}F label placement and reduce the time consuming effort of an entirely trial-and-error experimental approach.

MINOR ISSUES:

Could the authors comment on why they used MgCl₂ (instead of NaCl or KCl) in the C36m simulations to neutralise the box? Did the mobile Mg²⁺ ions cause any issues in the simulation, for example by overly strong binding to certain protein side chains (Glu, Asp)?

In line 672 (and probably elsewhere in the text), it should be "nm⁻²" instead of "nm²"

In line 693, it should probably say "or the tyrosine OH group"

(Remarks on code availability)

Version 1:

Reviewer comments:

Reviewer #1

(Remarks to the Author)

In the revised manuscript, the authors have addressed the minor issues in the original version. I recommend to publish on Nature Communications.

(Remarks on code availability)

Reviewer #3

(Remarks to the Author)

(Remarks on code availability)

We are satisfied with the changes made.

Response to reviewers

Reviewer comments (black)

Our response (blue)

Reviewer #1 (Remarks to the Author):

The article entitled “Rational design of ^{19}F NMR labelling sites to probe protein structure and interactions” by Julian et al. have systematically explored the strategic utilization of ring current effect of the aromatic residues to enhance the chemical shift dispersion of ^{19}F labels. ^{19}F NMR has been shown to have unique advantages to study complex protein targets, including multiple conformational dynamics in membrane proteins, protein folding of nascent chains on ribosome, protein-protein interaction in cells, etc. However, a significant limitation of highly sensitive and widely used $-\text{CF}_3$ based labels is the relatively small chemical shift dispersion, i.e., the extent of the response of its chemical shift to the environments, resulting in severe peak overlapping or inability to resolve the environment changes. Although the ring current effect has been found to be an important factor to modulate the ^{19}F chemical shift, it has never been systematically explored, and a reliable method to efficiently find the optimal ^{19}F labeling sites to monitor the protein behavior is still missing. To overcome this limitation, Julian et al. leverage the strength of AlphaFold and all-atom molecular dynamics to obtain a generalized geometric factor between the ^{19}F label and strategically chosen intrinsic or engineered aromatic residues and use the geometric factor to evaluate the potential secondary ^{19}F chemical shift changes. The method achieved an excellent success rate, and the authors have proved its power on several challenging applications. Particularly, the observation of the interaction between FLNa21 and migfilin and the folding thermodynamics of FLN5 in living cells is amazing. I believe that this method will greatly benefit the broad community of using ^{19}F NMR to study protein dynamics and protein-protein/ligand interactions. The manuscript is well prepared, however, I do have a few questions.

We are grateful to the reviewer for their highly positive comments about our manuscript.

1. While the comparison between Phe and His clearly showed the larger chemical shift changes, it would be great if the authors could use one representative case to compare Trp over Phe, although Trp is generally considered as less aromatic than Phe.

We thank the reviewer for raising this very good point. Introducing one, or both of Phe and His in the several cases investigated was favoured because of their smaller sizes of these sidechains relative to Tyr and Trp, resulting in these being observed in our MD design to be more favourably accommodated on the protein surface while minimising structural perturbations to other sidechain interactions. This was confirmed experimentally by the very small changes observed in the measured folding free energies arising from engineering these substitutions as shown in Supplementary Table 2. The focus on Phe arises, importantly, also through its stronger ring current than that of Tyr sidechain (due to the polar hydroxyl group attached to the aromatic ring of Tyr). However, in cases where Tyr is natively present and a nearby tmF then introduced, (exemplified by HRAS 32 tmF Tyr40, Supplementary Figure 5), a strong ring current could still be generated.

This raised point by the reviewer does necessitate some additional clarification regarding our design strategy and we have addressed this in the revised manuscript (lines 150-151 and 416-418).

2. In line 119, the chemical shifts are restricted only to solvent-exposed labeling sites. Are the analyses in Figure 2 restricted to solvent-exposed sites too? If it's true, the authors may need to give some explanations.

We thank the reviewer for pointing this out and have clarified that all variants in Figure 2 were restricted to solvent-exposed sites (line 146, Figure 2 legend and lines 937-938).

3. Figure 2e clearly shows correlation between the secondary ^{19}F chemical shifts and the geometric factor. For those with large deviations from the regression line, could they be rationalized by some significant factors like charge, polarity, or other effects?

While the overall correlation between the predicted ring currents and secondary chemical shifts is good, the reviewer correctly states that there are additional effects, including electrostatics, that may contribute to the secondary shift. We also noted this in lines 110-111.

The main determinant of the correlation and therefore the extent and number of outliers from the regression line, however, would appear to be the extent of the accuracy of the tfmF-aromatic sidechain interaction prediction. When comparing predictions from ColabFold, AF3, and MD (ff15ipq) (see Figure 2e), we observe an increase in the correlation coefficient from 0.53 to 0.70 (main text lines 212-232). This would indicate that the distances/angles of the tfmF-aromatic interactions are likely to be the most sensitive determinant of deviations from the regression line, as the structural environment and context (i.e., presence of charged and polar residues close to tfmF) are essentially identical in AlphaFold- and MD-based predictions. For example, the HRAS variant 157tfmF/153His deviates from the regression line by 0.14 ppm in the prediction using ColabFold but by as much as 0.51 ppm in the MD (ff15ipq) prediction (Supplementary Tables 3 and 5). Conversely, FLN5 655tfmF (contacting the wild-type residue Phe675) deviates from the regression line by 0.51 ppm using the ColabFold prediction but by only 0.27 ppm for the MD (ff15ipq) prediction (Supplementary Tables 3 and 5). We have further clarified this briefly in the Results section (lines 226-230).

For the MD (ff15ipq) predictions, the ¹⁹F chemical shift RMSD from the regression line is merely 0.19 ppm, which is similar to the ± 0.2 ppm chemical shift range around the random coil value of -61.82 ppm that we have observed for FLN5 variants without ring currents (Figure 1c). This indicates that these electrostatic and other effects contribute only to a small extent relative to ring currents; we have added a sentence to note this in the Results section (lines 226-230) of the revised manuscript.

While a detailed deconvolution and prediction/calculation of factors contributing to the chemical shift (including charge) would indeed be attractive, we have also observed that previous attempts, even with highly computationally expensive quantum mechanical calculations, have not been successful (ref. 28 and as stated in the introduction, i.e., lines 66-69 in the manuscript). Indeed, this highlights the strength of our approach, where the high success rates of our rational design approach (80-90%) demonstrates that the induced ring current can almost entirely account for the magnitude of the secondary chemical shift.

4. As an important method, some more details of experimental description would be very useful for readers and potential users. Especially considering so many expressions have been done for the incorporation of tfmF for this manuscript, could you mention in the Methods section a little about the protein yields?

The revised manuscript now includes a more detailed description of the general expression conditions and protein yields (lines 448-459).

Minor issue:

In Figure 6d, the colors of in-cell and lysate are indistinguishable, at least on my screen.

We thank the reviewer for pointing this out and have adapted this panel accordingly.

Reviewer #1 (Remarks on code availability):

I am not good at coding, so it's out of my ability to access the quality of the code.

Reviewer #2 (Remarks to the Author):

In this work, it is shown that ring currents affect the ^{19}F chemical shift of trifluoromethyl phenylalanine, making it a sensitive probe of protein conformation. The authors exploit this property and establish predictive computational tools that select artificial protein mutants in which the chemical shift of fmPhe is highly sensitive to small changes in local protein conformational states. This approach is then applied to investigate several different phenomena, including the identification of alternative conformational states, the characterisation of nascent protein folding intermediate structures on the ribosome, and probing protein-ligand and protein-protein interactions both in vitro and in bacterial cells. This is an important advancement that will greatly extend the applicability of ^{19}F NMR spectroscopy to investigate protein conformation and interactions, especially in challenging conditions such as with large complexes or in the cellular environment.

The work is elegantly designed, properly executed and clearly written, and the conclusions are fully supported by the experimental data.

Therefore, I recommend publication in Nature Communications, after few minor issues are addressed, mainly to improve the clarity and readability of text and figures.

We are grateful to the reviewer for their positive comments about our manuscript.

Minor points:

1) Although very pleasing to the eye, the choice of colours in some figures makes them hard to interpret: in Figure 1 panel c (top), shades of indigo-blue-turquoise are used to represent eighteen overlaid spectra. It is impossible to tell them apart even for a non-colourblind person. A similar problem is found in Figure 6d. Similarly, in Figure 1 f the colour coding chosen to represent the angle makes it hard to understand how variable the angle is in the MD simulation: angles between 0 and 45° almost look the same colour, even with a properly calibrated screen. The same issue applies to the same type of plot in all subsequent figures.

We thank the reviewer for pointing this out and have changed the colours of these figures accordingly, and have also improved the labelling within Figure 1c.

2) In all distance and angle plots of representative all-atoms MD simulations shown throughout the work, it would be useful to report the average distance and angle values calculated from the MD simulation, as they constitute the selection criterion for the mutations (if I understood correctly).

The average distances and angles (i.e., fraction of time spent in in-plane of the ring), which were indeed used as criteria for the mutations, are reported in Supplementary Tables 3-6 for all prediction methods and all 50 variants studied. The average geometric factors combining both distances and angles are also included in these tables.

3) "...from structures predicted by AlphaFold2 (AF2)/ColabFold..." How does ColabFold differ from AlphaFold2? As I understand, the former is an accessible implementation of the latter, so maybe this could be rephrased to make it clearer.

The reviewer is correct, and we have clarified this (line 155).

4) I had trouble understanding the construction of this sentence: "We further note that while labelling/residue pairs that are predicted with ColabFold at moderate to high confidence ($\text{pLDDT} > 70$ and > 90 , respectively) for both true and false predictions, AF3 has proportionally fewer moderate-to-high confidence false predictions" Is a verb missing in the first half?

We have clarified and rephrased this sentence in the revised manuscript (lines 207-210).

5) In Figure 3c, the dashed line corresponding to the random coil chemical shift is almost invisible.

We have increased the thickness and dash length to improve visibility.

6) Figure 3f might benefit from insets zooming in on the conformations of Phe and Tyr residues, which are otherwise drawn too small to be understood.

We agree with the reviewer and have added insets zooming in on the conformations of these residues in Figure 3f to improve visualisation of their conformations.

7) In Figure 4c, as GDP has two aromatic rings, with respect to which axis was the angle calculated?

We thank the reviewer for pointing out this minor omission and clarify this in the Methods (lines 621-622). For two-ring systems like GDP and Trp we used the centre of mass of all ring atoms as a proxy for the average position relative to the two rings.

8) In Figure 5 a and b, I cannot tell which chains belong to FLNa21 and which to migfilin. They have too similar shades of grey. Also, aromatic residues could be coloured consistent with the previous figures.

We have adjusted the shades of grey to better distinguish the FLNa21 and migfilin chains and coloured the sidechains consistent with previous figures.

Reviewer #3 (Remarks to the Author):

In the current work, Streit and coworkers use computational modeling to guide their labeling strategy for their experimental ^{19}F protein NMR methodology. They demonstrate high success rates of predicted ring current-induced chemical shift (CS) changes (relative to random coil values) using either AlphaFold models or MD simulations, with the latter being even more accurate in predicting CS changes. The demonstrated combined computational/experimental approach improves the CS dispersion and enables simple 1D NMR experiments to analyse structural features of alternative protein conformational states. This is shown with impressive examples ranging from ribosome-bound folding intermediates, and even in-cell NMR to probe protein interactions and folding thermodynamics.

The authors followed established state-of-the-art procedures to set up, run, and analyse their MD simulations. It is convincing that the authors used two current biomolecular force fields that are suitable for simulating also fluorinated amino acids, ff15ipq and C36m, which in most cases yielded consistent results. The simulation times of 1 microsecond per replicate (or even (much) longer for HemK, 20 microseconds per replicate) are state-of-the-art as well. Importantly, the authors also ran independent replicate simulations initiated from different rotamer states of the sidechains of interest, which (again) yielded consistent results. This shows the convergence of the simulations, and altogether it underlines the robustness of the simulation results. At the end of the day, most convincing is of course the experimental validation (or verification) of the predictions from the simulations. They can thus contribute to a rational design of strategic ^{19}F label placement and reduce the time consuming effort of an entirely trial-and-error experimental approach.

We thank the reviewer for their highly positive feedback.

MINOR ISSUES:

Could the authors comment on why they used MgCl_2 (instead of NaCl or KCl) in the C36m simulations to neutralise the box? Did the mobile Mg^{2+} ions cause any issues in the simulation, for example by overly strong binding to certain protein side chains (Glu, Asp)?

We thank the reviewer for raising this point. In this work since we are analysing neutral, aromatic sidechains where ions are not expected to bind directly, we expect the type of salt (NaCl vs MgCl_2) not to be a parameter that will impact the simulations significantly. Bolstering this assumption, we have generally also not observed (in previous works, refs. 8-9) the experimentally determined chemical shifts to be sensitive to the type of salts in the buffer, the Mg^{2+} concentration, nor total ionic strength. To clarify that the choice of salt is not critical for our approach, we have added a sentence in the Methods section (lines 522-525).

The use of two different salts arose because we had initiated the C36m and ff15ipq simulations described in the manuscript with the two force fields at different stages of the project, and where initially we had also used some experimental buffers that contained NaCl . As several extensive simulations had already been initiated prior to the salt switch, we had decided to keep the different salts within each force field for consistency.

We have also re-checked the interactions between negatively charged sidechains within the FLN5 655tfmF protein and Mg^{2+} ions. We find (left hand figure below) these to be predominantly transient (5-20%), rising to a maximum of up to 50%, (as would be typically expected). Such transient interactions would not influence ring current predictions and, indeed, results obtained with ff15ipq and C36m setups are consistent and both approaches/setups resulted in ~90% design success rates (Supplementary Figure 5g-h in the manuscript, and right hand side below):

Right: Average Mg^{2+} contact probability for all negatively charged amino acid sidechains in FLN5 655 *tfmF* (mean \pm s.e.m. from a 1 μs MD simulation).

Left: Correlation (and Pearson correlation coefficient, r) of average distances calculated for all variants simulated with both the ff15ipq and C36m force fields (mean \pm s.e.m. from 3 x 1 μs MD simulations per variant). These data included in Supplementary Tables 5 and 6 in the manuscript.

In line 672 (and probably elsewhere in the text), it should be "nm⁻²" instead of "nm²"

Many thanks for spotting this, we have now corrected this (lines 533 and 596 in the revised version).

In line 693, it should probably say "or the tyrosine OH group"

Now corrected (lines 617-618 in the revised version).